# Association between Peak Expiratory Flow Rate and Exposure Level to Indoor PM2.5 in Asthmatic Children, Using Data from the Escort Intervention Study

**DOI:** 10.3390/ijerph17207667

**Published:** 2020-10-21

**Authors:** Sungroul Kim, Jungeun Lee, Sujung Park, Guillaume Rudasingwa, Sangwoon Lee, Sol Yu, Dae Hyun Lim

**Affiliations:** 1Department of Environmental Sciences, Soonchunhyang University, Asan 31538, Korea; l01025263029@gmail.com (J.L.); psj57732398@gmail.com (S.P.); guillaumer1992@gmail.com (G.R.); solsol0914@gmail.com (S.Y.); 2Department of ICT Environmental Health System, Graduate School, Soonchunhyang University, Asan 31538, Korea; andrea0903@naver.com; 3Department of Pediatrics, School of Medicine, Inha University, Incheon 22332, Korea; dhyunlim@inha.ac.kr

**Keywords:** asthma, air purifier, indoor, PM2.5, PEFR, children, sensor

## Abstract

Various studies have indicated that particulate matter <2.5 μm (PM2.5) could cause adverse health effects on pulmonary functions in susceptible groups, especially asthmatic children. Although the impact of ambient PM2.5 on children’s lower respiratory health has been well-established, information regarding the associations between indoor PM2.5 levels and respiratory symptoms in asthmatic children is relatively limited. This randomized, crossover intervention study was conducted among 26 asthmatic children’s homes located in Incheon metropolitan city, Korea. We aimed to evaluate the effects of indoor PM2.5 on children’s peak expiratory flow rate (PEFR), with a daily intervention of air purifiers with filter on, compared with those groups with filter off. Children aged between 6–12 years diagnosed with asthma were enrolled and randomly allocated into two groups. During a crossover intervention period of seven weeks, we observed that, in the filter-on group, indoor PM2.5 levels significantly decreased by up to 43%. (*p* < 0.001). We also found that the daily or weekly unit (1 μg/m^3^) increase in indoor PM2.5 levels could significantly decrease PEFR by 0.2% (95% confidence interval (CI) = 0.1 to 0.5) or PEFR by 1.2% (95% CI = 0.1 to 2.7) in asthmatic children, respectively. The use of in-home air filtration could be considered as an intervention strategy for indoor air quality control in asthmatic children’s homes.

## 1. Introduction

The adverse health effects of fine particulate matter (PM < 2.5 μm in diameter (PM2.5)) are well-documented, especially on respiratory or cardiovascular health [1]. Korea has a high PM2.5 level during winter or spring due to its location, that is, on the downwind side of strong sources of anthropogenic emissions or domestic transportation emissions including precursors of PM2.5 [2]. As outdoor PM2.5 can infiltrate buildings [3], a considerable portion ends up inside the residential houses, mixing with indoor PM2.5 generated from indoor activities, including cooking and cleaning. Therefore, reducing indoor PM2.5 levels may result in substantial public health benefits [4,5].

Asthma is a chronic inflammatory airway disease, and environmental exposure to PM2.5 is a proven risk factor for its exacerbation in children [6]. Asthmatic children may be more vulnerable to PM2.5 exposure, compared with healthy children.

In a previous study, the symptom severity in asthmatic children increased with the degree of exposure to secondhand smoke, allergic risk factors [7], or outdoor air pollution [8]. Although the association of exposure to ambient air pollution with health impacts is well-known, information about the impact of indoor air quality with short-term intervention (using an air purifier) on respiratory health is still rare.

Observational panel studies with personal monitoring reported an adverse effect of the exposure to particulate matter on the peak expiratory flow rate (PEFR) in asthmatic children [9]. A few previous intervention studies showed that air purifiers could reduce indoor PM2.5 levels in asthmatic children’s homes or classrooms, with improvements in their nasal symptoms [10] or lung function [11], respectively, although outcomes were obtained from relatively small samples (*n* = 8 children per group) [10] and the target intervention area was the classroom rather than the home.

Previous studies also showed that the use of air cleaners in residences could reduce the levels of particles from various outdoor and indoor sources by 32–68% [12,13,14]. Time–activity studies have estimated that asthmatic children or adults tend to spend as much as 83% of their time indoors [15]. Therefore, indoor PM2.5 have gained great attention due to the large amount of time that asthma patients spend indoors, indicating that proper control of indoor air quality (IAQ) can likely contribute to improving asthmatic children’s symptoms, with the complete removal of personal exposure sources.

The health impact of short-term exposure to various indoor PM2.5 may differ among asthmatic children due to their activity patterns, baseline health status, or allergy pathology, including allergic inflammation and pulmonary physiology. Although recognizing the variation in asthma symptoms by patient may provide a basis for understanding the disease causality and developing exposure management strategies that contribute to enhanced protective regulation for asthmatic children while decreasing the likelihood of serious asthma outcomes [16], intervention studies assessing the association of lung function in asthmatic children and IAQ levels, considering the adjustment for seasonal effects and clustering within each patient, are still limited.

Recently, in Korea, an environmental health smart study with connectivity and remote sensing technologies (ESCORT) was designed to evaluate the associations of health benefits and reduction of exposure to environmental risk factors, such as PM2.5, via Internet of Things (IoT)-based systems, including indoor air purifiers, real-time indoor PM2.5 monitors, and sensor-based spirometry, in patients with asthma and those with atopic dermatitis in Seoul and its satellite cities. We evaluated the utility of a commercially available air purifier with a HEPA filter to reduce indoor PM2.5 levels and its corresponding effect on the PEFR of 26 asthmatic children recruited from ESCORT in a double-blinded randomized crossover intervention.

## 2. Materials and Methods

### 2.1. Study Design

The original study procedures were related to human participants and approved by the research ethics committee of Inha University Hospital (IRB No. 2018-07-007), and written informed consent was obtained from the legal guardians of all participants. In this randomized double-blind crossover intervention study, we placed one air filtration unit (Tower_XQ600, ATXH663-HWK, Winix, Korea) in the bedroom or main living space of each participant’s home for two separate intervention periods, which lasted for three weeks each. These intervention periods had a washout period of at least one week (Figure 1) in between. To adjust for seasonal effects, we conducted measurements of IAQ and lung function within one season, that is, the fall of 2018 (September to October).

Participants served as their own control subjects. This study design could provide the strongest empirical evidence of treatment efficacy, and minimize selection bias and confounders due to unequal distribution of demographic, socioeconomic, or seasonal factors. It is also known that statistical test results of significance are readily interpretable [17].

During each intervention period, participants were exposed to two scenarios in random order: unfiltered air (control: no filter installed in filtration unit) and filtered air (experiment). Every air purifier used a brand new high efficiency particulate air (HEPA) filter (CAF-E0S4, Winix, Korea) for the experiment in phase 1 (first three weeks) or phase 2 (second three weeks), although the manufacturer mentioned that we can use the HEPA filter for at least six months in the home environment. All air purifiers exhibited an identical outward appearance regardless of filtration status. This filter system has a certified clean air delivery rate of 98 ft^3^/min, and it covers approximately 628 ft^2^.

### 2.2. Study Population

We used data collected from 26 asthmatic children who had been registered with the Environmental Health Research Center for Allergy and Respiratory Diseases, Inha University Hospital, Korea. All participants were diagnosed with mild asthma by a physician in the hospital according to the Global Initiative for Asthma guidelines [18].

Patients with mild asthma who had an episode of asthma exacerbation <2 times in the past 12 months, requiring an increase in medication, including systemic steroids, were recruited from Inha Hospital. We included only those participants with a PEFR range from 100 to 500 L/min. Participants were residents of Incheon metropolitan city (population, 10.3 million). A baseline clinical test was performed at their visit according to an international guideline [19]. For validation purposes, we collected data on the history of medications, such as anti-asthmatics, within three weeks prior to the study. Children born prematurely or those with an immune disorder were excluded from our study. We also collected questionnaires at baseline to understand their demographic and socioeconomic characteristics.

### 2.3. Measurement of PEFR and Fractional Exhaled Nitric Oxide (FeNO)

Daily PEFR was recorded using a SmartOne Spirometer (MIR Medical International Research, Roma, Italy). Our field manager explained the maneuver and demonstrated how to use SmartOne to children and their guardians before the actual recording. Each child was asked to take a deep breath and then blow into the peak flow meter as hard and quickly as possible; they were given two trial runs each and encouraged to blow harder each time. Daily maximum PEFRs were selected from the morning and afternoon measurements. A tight seal was maintained between the lips and the mouthpiece. According to the manufacturer’s document [20], SmartOne had an accuracy test outcome that complied with the American Thoracic Society (ATS) and European Respiratory Society (ERS) 2005 standards, the International Organization for Standardization (ISO) 26782 standard (for spirometry parameters), and the ISO 23747 standard (only for the peak flow parameter). The 2009 ISO 26782 standard and 2005 ATS/ERS Spirometry Statement agree that the accuracy of both volume- and flow-type spirometers should be checked [21].

Fractional Exhaled nitric oxide (FeNO) measurements were performed using a NIOX MINO^®^ (Aerocrine, Solna, Sweden) [22]. The asthmatic children sat on a chair wearing a nose plug, and they inhaled nitric oxide-free air to their maximum lung capacity. Children exhaled 50 mL/s while biting a mouthpiece and looking at a monitor. The test was performed once at baseline and repeated during the hospital visit (last day of each phase). We also obtained the Immunoglobulin E (IgE) level measured from the peripheral blood sample collected at baseline [23]. We used FeNO or IgE data for only descriptive analysis because only their baseline data were available for 26 children, while we had daily measurement data for PEFR and IAQ.

### 2.4. Indoor Air Pollution Data

IAQ data, including PM2.5, carbon dioxide (CO_2_), temperature, and relative humidity (RH), were obtained from ESCORTAIR and PurpleAir (PA) (PurpleAir, LLC, Draper, Utah, USA) with a 2-min interval through an on-board laser light scattering sensor installed at the main living space of the participants. Subsequently, to match our daily symptom data, we calculated the daily mean values of IAQ. We previously evaluated the performance of our real-time monitors and compared them with the US federal equivalent method (FEM), and we published outcomes in a separate study [24]. Our previous study found that the performance of this portable device was comparable to that of a research-grade expensive portable monitor [24].

We measured PM2.5 levels for seven weeks continuously in each participant’s home. As a routine calibration procedure, prior to sending our device to participants for data collection, devices were gathered and operated simultaneously for two days in a test laboratory in Soonchunhyang University with a reference device, GRIMM (MODEL 11-D, GRIMM Aerosol Technik Ainring GmbH & Co. KG, Ainring, Germany) Precision, which was calculated with relative standard deviation, was <15%.

The short-term performances of our devices were further validated by passing a national certification guideline prepared by the Special Act on the Reduction and Management of Fine Dust, Korea, amended by Act No. 16303 which came into force on 26 March 2019 [25]. According to the guideline, we compared ambient PM2.5 levels measured using our PurpleAir_K, (Purple Air, LLC, Draper, UT, USA; with a correction algorithm developed by ESCORTAIR, Asan, Korea) (PAs) and reference filter samplers prepared by the nationally authorized test site. In a four-week outdoor certification test period, we had cold (−7 to 8 °C as daily mean value) and rainy days (25% of test period). The time-averaged ambient hourly PM2.5 levels were from 5 to 40 μg/m^3^. A linear model related to the national site reported by a tapered element oscillating microbalance and PA monitors had a coefficient of determination (R2) of 99%. Our final hourly results after application of field correction were within 14% of the reference for 100% of samples. Since our main goal was the evaluation of PM2.5 levels depending on daily activities, we used time-series PM2.5 level data and were unable to characterize PM2.5 components.

### 2.5. Data Analysis

Descriptive statistics (medians and interquartile ranges (IQR)) were used to describe the study population. We determined differences in IAQ and PEFR levels according to the study groups (experiment and control), using the Chi square test or Mann–Whitney U test. Statistical significance was based on <0.05. For Mann–Whitney U test, we assumed that all observations of PM2.5 in the control and experiment groups were independent of each other. The PM2.5 measurements were ordinal. There was no relationship between weekly PM2.5 measurements of the control group and those of the experiment group. In addition, there was no relationship between weekly PM2.5 values in both groups.

To evaluate the impact of the use of air purifiers on changes in IAQ and consecutively on changes in PEFR, we used a multilevel model procedure to account for the clustered data (IAQ and PEFR). A random intercept model was used, and the coefficients and standard errors were evaluated under restricted maximum likelihood estimation with unstructured autocorrelation. In the development stage of our final multilevel model, we used only a patient’s data if he/she provided data over the intervention period, especially including at least four days’ data at the third week of each intervention period (control and experiment) with our assumption of a full change of IAQ by the intervention.

We assumed that there was a two-level structure within the data: episode level (level 1) and individual patient level (level 2). The episode level included PEFR results with the level of exposure to IAQ. The individual level included sex and age. Using the combined level 1 and 2 datasets, multilevel modeling provides a way to analyze the degree to which the effects of episode level variables vary systematically or randomly, as a function of the individual (level 2) variables [26].

We assumed that the intercepts for different patients were normally distributed. To estimate the dependency of the measurements of the lung function of each patient, we calculated the intraclass correlation (ICC) coefficient, defined as the variance between patients divided by the total variance (summation of the variance between patients and variance within a patient); ICC increased as variance decreased [26].

Log transformation was conducted for the dependent variable, PEFR, in the regression analysis to account for the right-skewed distribution. The geometric mean ratio of PEFR was calculated to estimate its association with IAQ change according to the use of air purifiers. We adjusted for sex, age, and phase type, and sensitivity analyses were conducted. Instead of daily measurement, we used weekly median values of the measurement and conducted the modeling analysis. All analyses were performed using the SAS 9.3 package (SAS Institute, Cary, NC, USA).

## 3. Results

The summary of the baseline demographic information of participants is presented in Table 1. The median (IQR) age did not differ according to baseline intervention status (Mann–Whitney U test *p*-value = 0.7734 with 8.5 (7.0–9.0) years for the control (*n* = 14) and 9.5 (6.0–11.0) years for the experiment group (*n* = 12)). The median (IQR) body mass index (BMI) was 17.4 (15.3–23.0) kg/m^2^ for the control group and 18.1 (15.8–23.6) kg/m^2^ for the experiment group (*p* =0.51). In addition, the IgE levels were not different (*p*-value = 0.3418): 131.9 (16.1–527.9) IU/mL and 259.3 (119.8–462.8) IU/mL for the experiment and control groups, respectively.

There was no significant difference in FeNO level (*p*-value = 0.6218), where the median FeNO level (IQR) for the control group was 20.5 (18.0–28.0) ppb and that for the experiment group was 16.0 (10.0–32.0) ppb. Similar to baseline BMI, IgE, or FeNO, the median (IQR) PEFR values between the two groups were not significantly different: 426.0 (395.0–446.5) L/min for the control group and 393.0 (222.0–402.0) L/min for the experiment group (*p* = 0.3407).

At baseline, the daily median (IQR) indoor PM2.5 level was similar between the control and experiment groups (*p* = 0.4307): 18.9 (17.3–19.4) μg/m^3^ for the control group and 17.0 (12.0–19.9) μg/m^3^ for the experiment group. The median (IQR) of CO_2_ was also similar by intervention status (*p*-value = 0.8606), with the control and experiment groups accounting for 501.6 (488.4–627.2) ppm and 676.5 (444.2–908.8) ppm, respectively. The indoor median (IQR) temperature and RH was not significantly different (*p* = 0.4395 or 0.8483, respectively) between the groups (Table 2).

As shown in Table 3, after deploying filters inside the participants’ homes, the indoor PM2.5 level decreased by approximately 50% (*p*-value < 0.001) in three weeks of observation per phase. However, CO_2_, temperature, and relative humidity did not change significantly. Additionally, the distribution of PEFR values was not significantly different between the control and experiment groups (*p* = 0.6231).

Figure 2 shows the distributions of IAQ and PEFR measurements by week and phase. The indoor PM2.5 levels were significantly lower (*p* < 0.05) in the experiment (filtered) group than in the control (unfiltered) group regardless of the week or phase. However, in the case of PEFR, there was some decrease, especially in the first phase; however, this was not statistically significant.

As seen in Figure 3, the distributions of weekly differences in PEFR or PM2.5 values from their overall median value, obtained over the entire study period, demonstrated lower indoor PM2.5 values in the experiment group for both phases. PEFR levels were similar or slightly lower in the experiment group than in the control group.

Using multilevel modeling, we further evaluated the association between IAQ and PEFR by considering the clustering of data by patient. Table 4 shows the multivariate geometric means and 95% confidence of the IAQ and asthmatic children’s PEFR obtained from the model after adjusting for potential covariates such as sex, age, and phase of intervention. According to Table 4, multilevel model 1 showed that decreased PEFR in asthmatic children (geometric mean ratio, GMR = 0.2%, 95% CI = 0.1–0.5% was significantly associated with unit increase in indoor PM2.5 levels after adjustment for age, sex, and phase. When we used weekly median values for all variables, we observed that PEFR possibly decreased by 1% (95% CI = 0.1–2.7%) if their weekly indoor PM2.5 level increased by 1 μg/m^3^.

## 4. Discussion

This intervention study showed that the use of high efficiency particulate air (HEPA) purifiers decreased indoor PM2.5 levels by up to 43% at the asthmatic children’s homes in terms of median indoor PM2.5 levels between the control group (air purifier without a filter) and the experiment group (air purifier with a filter) over each of the three-week intervention periods. Our study results were similar to the results of a number of other studies that evaluated the effect of air filters in residential settings and showed consistent reductions in indoor PM2.5 with HEPA filter deployment [13,27,28]. HEPA filter use was not significantly associated with a reduction in CO_2_ level, which may partly be due to the fact that ventilation and occupancy of residential settings were not altered by filter operation [29].

In terms of the direction of impact regarding the association between PM2.5 level and PEFR change, we found consistency between our outcome and previous US or Japanese studies. In a study by O’Connor et al. [30], which was conducted in a two-week period in 2008, using over two years’ of pulmonary function data from 861 asthmatic children (mostly Hispanic and black), aged ≥7 years, in several inner cities in the USA, they found a 1.1% (95% CI, −0.56 to −1.65) decrease in PEFR per short-term increase of 13.2 μg/m^3^ in PM2.5 as a five-day average value from a single pollution model, while they reported a 0.25% (95% CI, −0.88 to 0.38) decrease in PEFR based on the same rate of increase in PM2.5 from their three pollutant models. Their associations were obtained with a short-term increase in ambient air pollutant concentration, which were even below the US national ambient air quality standards.

In our study, we found that the difference in indoor median PM2.5 level between the control and experiment groups was 6.5 μg/m^3^, and decreased PEFR in asthmatic children (0.2%, 95% CI = 0.1–0.5%) was significantly associated with unit increase in indoor PM2.5 level, after adjusting for other IAQ or demographic variables with a random intercept for each asthmatic children in our multilevel model. Our median (IQR) indoor PM2.5 levels were also even below the WHO’s current 24-hour exposure guideline (25 μg/m^3^) [31].

A panel study conducted by Odajima et al. in Japan [8] also found that, in warmer periods (April through September), morning peak expiratory flow (PEF) declined by approximately 0.7 L/min (*p* < 0.05) by 10 μg/m^3^ differences in 3 h concentration of outdoor suspended particulate matter (SPM) (any particle collected with an upper 100% cutoff point of a 10µm aerodynamic diameter). They did not find any robust association between SPM and morning/evening PEF during the colder months in their study area (Fukuoka, Japan).

The discrepancies between studies regarding the margin of association were not surprising because US and Japanese studies used estimates of outdoor PM2.5, whereas our study was conducted using indoor PM2.5. As mentioned earlier, because asthmatic children tend to spend more than 80% of their daily time indoors, although the margin of IAQ change is smaller due to relatively longer duration of exposure indoors than the exposure outdoors, the impact on PEFR can be larger. For better understanding, the causal components of PEFR reduction and susceptible groups of indoor PM, compared to outdoor PM, should be studied [32].

Although we found a potential association between the exposure to indoor PM2.5 and PEFR change, our study should be interpreted with caution. Indoor air pollution levels may not adequately reflect the total exposure rate because, despite the deployment of filters inside children’s homes, it is uncertain whether all their time was spent indoors where the PM2.5 levels significantly decreased. Therefore, the associations between PM2.5 and PEFR in this study may have been due to misclassification of the children’s actual exposure duration. Moreover, a relatively small proportion of participants resulting in an insufficient detection power may have contributed to the association between the PM2.5 level and PEFR due to the underestimation of air pollution influence, as shown by other studies [33]. In this study, we thought about calculating “average daily dose” considering their daily activity pattern and daily exposure duration, by providing them with an activity pattern diary to develop micro-environmental models. However, we had information on activity patterns from only 15 of the 26 children. Thus, to obtain appropriate statistical power, we decided to use indoor PM2.5 level rather than average daily dose. Our study outcomes of average daily doses of PM2.5 for major activity patterns from those 15 children are presented in our previous study [34].

In this study, during the statistical analyses process, we intended to adjust the effect of FeNO, but due to the lack of data (*n* = 1 per phase over three weeks), we could not conduct further analysis. A future study may be useful to show the effect of indoor PM2.5 change after adjusting for weekly FeNO levels. In addition, although we tried to minimize the effect of potential confounders by conducting our study within one season with an intervention study design that allowed each participant to serve as their own control subjects, information on the medication taken by asthmatic children, protecting against the inflammatory effects of PM2.5 [35], was not obtained on a daily basis. A future study evaluating the degree of interaction between medication use and indoor exposure to PM2.5 is required.

Nevertheless, this study showed the effectiveness of a commercially available air purifier with a HEPA filter, in reducing indoor PM2.5 exposures over a three-week study period per phase, demonstrating potentially relevant protective respiratory health outcomes among asthmatic children.

## 5. Conclusions

In this study, we found that a daily or weekly unit increase in indoor PM2.5 level could significantly decrease PEFR by 0.2% (95% CI = 0.1 to 0.5) or by 1.2% (95% CI = 0.1 to 2.7) in asthmatic children, respectively. From a conservative public health point of view, the use of in-home HEPA filter air cleaners should be considered as a comprehensive intervention strategy for IAQ in asthmatic children’s homes.

## Figures and Tables

**Figure 1 ijerph-17-07667-f001:**
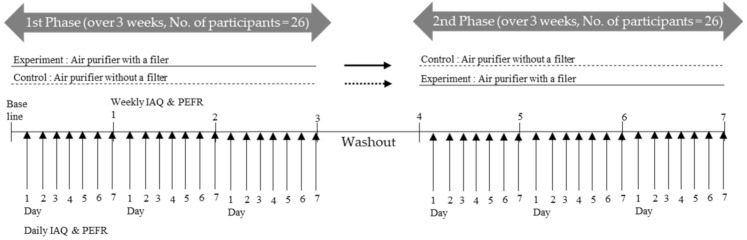
Randomized double-blind crossover intervention study design used in this study. Informed consent was approved by the research ethics committee of Inha Hospital. IAQ: indoor air quality; PEFR: peak expiratory flow rate.

**Figure 2 ijerph-17-07667-f002:**
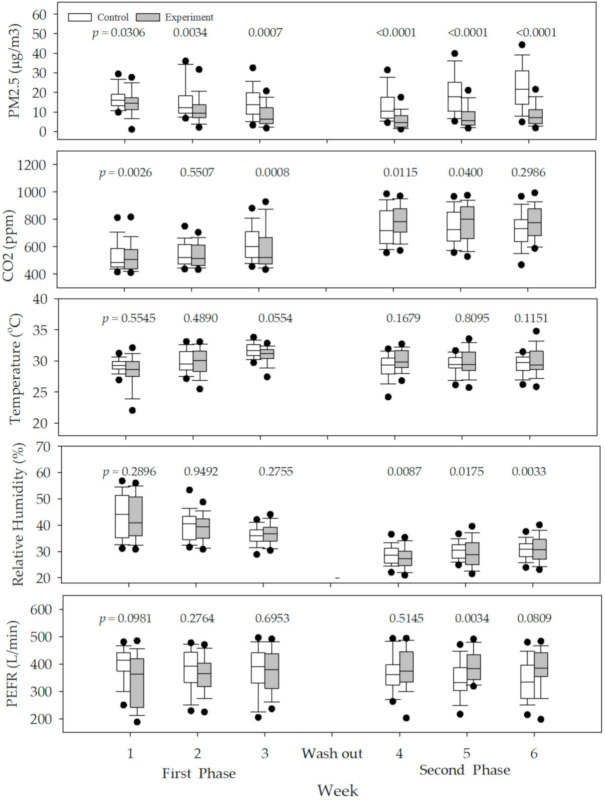
Distributions of daily indoor particulate matter <2.5 μm (PM2.5), temperature, relative humidity, CO_2_, and peak expiratory flow rate (PEFR) measurements per week during the intervention period. Significant difference of the distribution between the two groups was evaluated using Mann–Whitney U test. We provided *p*-values on the graphs.

**Figure 3 ijerph-17-07667-f003:**
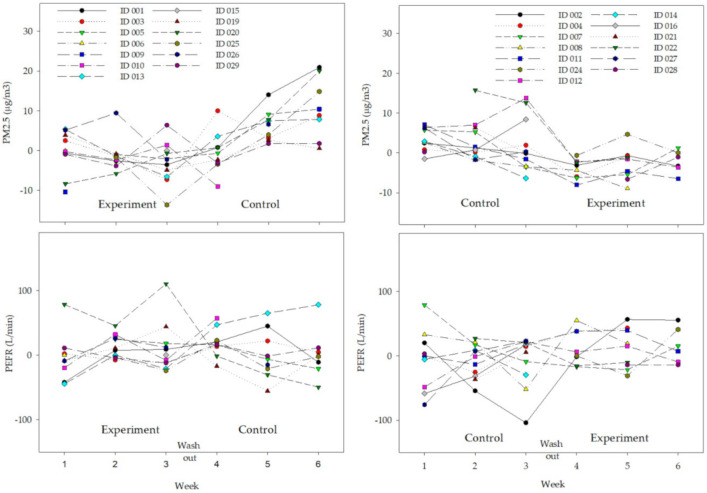
The distribution of person-specific weekly difference of PEFR or PM2.5 values from their median value, obtained over the study period, by intervention group.

**Table 1 ijerph-17-07667-t001:** Baseline (first day) demographic and clinical information (median IQR).

Variable	Control (*n* = 13)	Experiment (*n* = 13)	*p*-Value *
Sex (F, %)	15.3	7.6	0.3562
Age (years)	8.5 (7.0–9.0)	9.5 (6.0–11.0)	0.7734
BMI (kg/m^2^)	17.4 (15.3–23.0)	18.1 (15.8–23.6)	0.5714
BMI percentile	78.6 (48.8–94.6)	85.6 (61.2–96.9)	0.4558
Height (cm)	130.4 (128.2–139.1)	134.5 (121.7–146.7)	0.9385
Weight (kg)	29.3 (26.6–41.6)	33.5 (23.9–49.5)	0.8571
IgE (IU/mL)	131.9 (16.1–527.9)	259.3 (119.8–462.8)	0.3418
FeNO (ppb)	20.5 (18.0–28.0)	16.0 (10.0–32.0)	0.6218
PEFR (L/min)	426.0 (395.0–446.5)	393.0 (222.0–402.0)	0.3407

Note. * *p*-value from Chi-square test (sex) or Mann–Whitney U test (others); BMI: body mass index; IgE: Immunoglobulin E; FeNO: Fractional Exhaled nitric oxide; PEFR: peak expiratory flow rate.

**Table 2 ijerph-17-07667-t002:** Summary (median interquartile range, IQR) of baseline indoor air-quality information.

Variable	Control (*n* = 13)	Experiment (*n* = 13)	*p*-Value *
PM2.5 (μg/m^3^)	18.9 (17.3–19.4)	17.0 (12.0–19.9)	0.4307
CO_2_ (ppm)	501.6 (488.4–627.2)	676.5 (444.2–908.8)	0.8606
Temperature (°C)	29.0 (28.1–29.4)	29.0 (27.7–29.3)	0.4395
Relative humidity (%)	54.7 (53.8–57.3)	55.7 (53.5–56.1)	0.8438

Note. * *p*-value from the Mann–Whitney U test.

**Table 3 ijerph-17-07667-t003:** Summary (median (IQR) of indoor air quality between control and experiment groups for phase 1 and 2.

Variable	Control (Filter Off, *n* = 247)	Experiment (Filter On, *n* = 224)	*p*-Value *	*p*-Value **
PM2.5 (μg/m^3^)	15.3 (10.2–20.5)	8.8 (4.5–14.2)	<0.001	0.0001
CO_2_ (ppm)	676.7 (533.9–909.6)	639.2 (487.0–802.2)	0.3026	0.2559
Temperature (°C)	29.4 (28.6–30.6)	29.7 (28.8–31.7)	0.6050	0.0088
Relative humidity (%)	33.8 (30.7–38.8)	35.3 (28.6–40.2)	0.8066	0.2800
PEFR(L/min)	372.0 (319.0–430.0)	378.0 (328.0–417.0)	0.6231	0.8804

Note. * *p*-value from Mann–Whitney U test; ** Wilcoxon signed rank sum test for the difference in weekly median value by id and order of phase (1st, 2nd and 3rd weekly median values of phase 1 and phase 2).

**Table 4 ijerph-17-07667-t004:** Multivariate geometric means and 95% confidence intervals of indoor air quality and asthmatic children’s PEFR.

Variable		Multivariate 1 W/Daily Data (−log Likelihood = 154, AIC = 150)			Multivariate 2 W/Weekly Median Data (−log Likelihood = 897, AIC = 899)		
		GMR	95% CI		GMR	95% CI	
			Lower	Upper		Lower	Upper
PM2.5 (μg/m^3^)		0.998	0.995	0.999	0.988	0.973	0.999
CO_2_ (ppm)		1.0001	0.9999	1.0003	0.9999	0.9993	1.0004
Temp (°C)		1.006	0.994	1.019	0.993	0.945	1.041
Humidity (%)		0.999	0.995	1.003	0.991	0.975	1.007
Age (years)		1.067	1.032	1.104	1.028	0.983	1.068
Sex	Female (Ref)						
	Male	1.057	0.902	1.238	0.968	0.776	1.160
Group	Phase 1 (Ref)						
	Phase 2	0.949	0.885	1.019	0.897	0.677	1.117

Note. GMR: geometric mean ratio; 95% CI: 95% confidence interval.

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
