# Peer review of "Association between Peak Expiratory Flow Rate and Exposure Level to Indoor PM2.5 in Asthmatic Children, Using Data from the Escort Intervention Study"

_ijerph, 2020, doi:10.3390/ijerph17207667_

Round 1

Reviewer 1 Report

The authors have improved the manuscript to some extent. Some additional comments to be addressed or considered:

  1. Please check the English writing again.
  2. Figure 3: PM2.5 concentration had negative values. Why?
  3. As stated in Line 249, the association between PEFR and indoor PM2.5 levels was significant. What was the p-value? A little bit surprising to me as there was no significant difference in PEFR between the control and experiment groups.
  4. The conclusion section was bizarre: it did not even mention PEFR, which was the focus of the study.
  5. I would suggest using the abbreviation PM2.5 in the title instead of the "particulate matter less than 2.5 micrometers in diameter".

Author Response

The authors have improved the manuscript to some extent. Some additional comments to be addressed or considered:

  1. Please check the English writing again.

Thank you for your suggestion. We have revised the English writing according to your suggestion.

  1. Figure 3: PM2.5 concentration had negative values. Why?

To provide a useful interpretation, in Figure 3, we provided the difference in PM2.5 (or PEFR) value from their overall median value, over the study period (grand median value).

Therefore, weekly median PM2.5 (or PEFR) value was lower than the grand median value, which led to the occurrence negative values.

Reference: Kim S, Bukcley T, Dominici F, Concentrations of vehicle-related air pollutants in an urban parking garage, Environmental Research 105 (2007) 291–299.

To provide clarification, we included the following information in the title of Figure 3 and main body of text as seen below:

Figure 3. The distribution of person-specific weekly difference of PEFR or PM2.5 value from their median value, obtained over the entire study period, by intervention group.

(Line 238 ~241)

As seen in Figure 3, the distributions of weekly differences in PEFR or PM2.5 values from their overall median value, obtained over the entire study period, demonstrated lower indoor PM2.5 value in the experimental group for both phases. PEFR levels were similar or slightly lower in the experimental group than in the control group.

  1. As stated in Line 249, the association between PEFR and indoor PM2.5 levels was significant. What was the p-value? A little bit surprising to me as there was no significant difference in PEFR between the control and experiment groups.

The p-value was 0.04, which indicated a significant association between PEFR and indoor PM2.5 levels. As mentioned in other literatures, either P values or confidence intervals can be used to determine whether the results are statistically significant.

In our study, to demonstrate the statistical significance in our results, we provided the 95% confidence intervals for the association (geometric mean, GM=0.998, 95%CI=0.995–0.999) in the results section as well as the discussion section. This confidence interval for GM (0.998) indicated that the regression coefficients might be between 0.995 and 0.999 with 95% confidence.

References: 1. https://en.wikipedia.org/wiki/Simple_linear_regression

  1. Kim S et al., Determinants of Hair Nicotine Concentrations in Nonsmoking Women and Children: A Multicountry Study of Secondhand Smoke Exposure in Homes, Cancer Epidemiology, Biomarkers & Prevention, 2009;18(12):3407–14 DOI:10.1158/1055-9965.EPI-09-0337 

Line 251

According to Table 4, multilevel model 1 showed that decreased PEFR in asthmatic children (GM=0.998, 95%CI=0.995–0.999) was significantly associated with unit increase in indoor PM2.5 levels after adjustment for age, sex, and phase.

In this study, we found a large change in indoor PM2.5 levels over our intervention period (between the control and experiment periods, 3 weeks each), and a statistically significant association of 0.2% (95% CI, 0.1 ~ 0.5%) decrease in PEFR per short-term (daily) increase in indoor PM2.5 level by 1 μg/m3.

We agree with the reviewer that, compared with the degree of median indoor PM2.5 concentration change (6.5 μg/m3) between the control and experiment groups, that of PEFR was not so dramatic. However, as seen in Figure 3, we could observe an increase in PEFR trend when PM2.5 level reduced and a decrease in PEFR trend when PM2.5 level rose, in each participants’ data. Similar results were found in previous study results, as we mentioned in the discussion section.

Line 269

According to O’Connor et al.’s study [30], which was conducted in a 2-week period, using pulmonary function data from 861 asthmatic children (mostly Hispanic and black), aged ≥7 years, for over 2 years, in several inner cities in the USA in 2008, they found a 1.1% (95% CI, -0.56 ~ -1.65) decrease in PEFR per short-term increase of 13.2 μg/m3 as a 5-day average PM2.5 value from a single pollution model.

  1. The conclusion section was bizarre: it did not even mention PEFR, which was the focus of the study.

Thank you, we revised our study conclusion as suggested.

In this study, we found that the daily or weekly unit increase in indoor PM2.5 level could significantly decrease PEFR by 0.2 (95% confidence interval [CI] 0.1 ~ 0.5)% or by 1.2 (95 % CI 0.1 ~ 2.7)% in asthmatic children, respectively. From a conservative public health point of view, the use of in-home HEPA filter air cleaners should be considered as a comprehensive intervention strategy for IAQ in asthmatic children’s homes.

  1. I would suggest using the abbreviation PM2.5 in the title instead of the "particulate matter less than 2.5 micrometers in diameter

Thank you for your suggestion. We have used the “PM2.5” in the title, instead of the "particulate matter less than 2.5 micrometers in diameter”

Reviewer 2 Report

I have read the article by Kim et al. with great interest. This is a very nicely conducted trial and a well-written paper.

Comments:

  • Was FENO measured once or in duplicates? It is suggested to do multiple tests and use their average, however I understand that due to the cost of reagents it is not always the case and only one measurement is used. Please, clarify in the article.
  • Higher FENO makes asthmatic people susceptible for bronchoprovocation positivity (i.e. mannitol, methacholine, exercise, etc.). I wonder if this is the case for PM2.5. What happens if you adjust your multivariate model for FENO as well?

Author Response

Comments and Suggestions for Authors

Comments:

ï¾· Was FENO measured once or in duplicates? It is suggested to do multiple tests and use their average, however I understand that due to the cost of reagents it is not always the case and only one measurement is used. Please, clarify in the article.

For FeNO measurement, Niox Mino products were used and the test was performed once at baseline and repeated during hospital visit as stated in the original manuscript. We measured the FeNO once per visit because of its high measurement cost (38 USD per time). 

We used FeNO or IGE data for only descriptive analysis because only their baseline data were available for 26 children, while we had daily measurement data for PEFR and IAQ.

We clarified the statement as seen below:

Line 132

FeNO measurements were performed using a NIOX MINO® (Aerocrine, Solna, Sweden) [22]. The asthmatic children sat on a chair wearing a nose plug, and they inhaled nitric oxide-free air to their maximum lung capacity. Children exhaled 50 mL/s while biting a mouthpiece and looking at a monitor. The test was performed once at baseline and repeated in during  hospital visit (last day of each phase). 

ï¾· Higher FENO makes asthmatic people susceptible for bronchoprovocation positivity (i.e. mannitol, methacholine, exercise, etc.). I wonder if this is the case for PM2.5. What happens if you adjust your multivariate model for FENO as well?

Thank you for mentioning the bronchial provocation test (BPT). We  performed a provocholine BPT when a patient was enrolled at baseline. However, the test was not conducted during this study period because our main response variable was PEFR.

FeNO tests were performed only on the last day of each phase, and we could not perform any further analysis using them in our multilevel-multivariate model. 

In future, we will examine the change in BPT outcome by PM2.5, or the effect of PM2.5 on FeNO, using clustered data.

We described these issues as our study limitation in our discussion section:

In this study, during the statistical analyses process, we intended to adjust the effect of FENO, but due to the lack of data (n=1 per phase over 3 weeks), we could not conduct further analysis. A future study may be useful to show the effect of indoor PM2.5 change after adjusting for weekly FENO levels.

Reviewer 3 Report

1) Abstract: it would be good to include in the abstract the location the study was conducted. This is highly relevant to the exposure under study here (indoor PM2.5).

2) Abstract: line 25: with ‘unit increase’ do the authors mean an increase of 1 ug/m3? It would be good to be specific here.

3) Figure 1: This figure seems to suggest that daily indoor air quality measurements (IAQ) and peak flow measurements (PEFR) were only done in the first week of each 3-week phase. Surely this is not correct?

4) results line 199-210: it does not seem necessary to repeat here in detail what is already in the table.

5) results line 224: it states here that CO2 did not change statistically significantly. However, the p-value of 0.0026 indicates that it did.

6) The Mann-Whitney U test is appropriate if all the observations from both groups are independent of each other. This assumption is however not met in Table 3. Here each daily indoor air quality and PEFR measure taken during the intervention is compared with those taken during the non-intervention period. However, the daily indoor air quality and PEFR measurements taken from the same house and same individual, are of course not independent of each other. The authors may want to consider other tests here (e.g. paired t-test using the individuals’ medians over the periods).

7) Table 4: I would expect each participant to serve as their own control in this analysis, but that does seem not to be the case. I note that the discussion states on line 307-308 that “our… study design allowed that each participant served as their own control”. This is however not the case in the presented analyses.

8) discussion line 258: “decreased indoor PM2.5 by up to 43%”. What is meant here? Did PM2.5 decrease by 43% on average, or does the 43% relate to the largest decrease observed? The same goes for line 273: “median PM2.5 could decrease as much as 6.5 ug/m3”. What is meant exactly here?

9) line 303: here it states the study included 30 children. Throughout the manuscript a total of 26 children is mentioned. What happened here? Similarly, the number of measurements listed in Table 3 suggest that only about 50% of the daily measurements were available. What happened here? The exact number of participants and number of measurements (n aimed for and n available for analysis) needs to be described.

Author Response

Comments and Suggestions for Authors

1)   Abstract: it would be good to include in the abstract, the location the study was conducted. This is highly relevant to the exposure under study here (indoor PM2.5).

Thank you for your suggestion. We included the study location accordingly.

“This randomized, crossover intervention study was conducted in 26 asthmatic children’s homes located in Incheon metropolitan city, South Korea.” 

2)   Abstract: line 25: with ‘unit increase’ do the authors mean an increase of 1 ug/m3? It would be good to be specific here.

Thank you. We have specified as suggested.

“We also found that the daily or weekly unit (1 g/m3) increase in indoor PM2.5 level could significantly decrease PEFR by 0.2 (95% confidence interval [CI] 0.1 ~ 0.5)% or PEFR by 1.2 (95%CI 0.1 ~ 2.7)% in asthmatic children, respectively.”

3)   Figure 1: This figure seems to suggest that daily indoor air quality measurements (IAQ) and peak flow measurements (PEFR) were only done in the first week of each 3-week phase. Surely, this is not correct?

We have provided clarification accordingly; we updated our Figure 1 as seen below.

After

Before

4)   results line 199-210: it does not seem necessary to repeat here in detail what is already in the table.

Thank you for your suggestion. Although the information is presented in the tables, we provided simple statements to help the reader’s understanding.

5)   results line 224: it states that CO2 did not change statistically significantly. However, a p-value of 0.0026 indicates that it did.

Thank you for pointing this out. We apologize for this mistake. To provide clarification, we revised the error from 0.0026 to 0.3026 in Table 3.

6)   The Mann-Whitney U test is appropriate if all the observations from both groups are independent of each other. This assumption is however not met in Table 3. Here each daily indoor air quality and PEFR measure taken during the intervention is compared with those taken during the non-intervention period. However, the daily indoor air quality and PEFR measurements taken from the same house and same individual, are of course not independent of each other. The authors may want to consider other tests here (e.g. paired t-test using the individuals’ medians over the periods).

We appreciate your idea of testing the difference in medians over the period. Hence, we included the p value obtained from the Wilcoxon signed rank sum test as a nonparametric alternative to the paired t-test. 

Since the distributions of PM2.5 (or PEFR) of the control or experimental group were not normal, we could not apply any type of T-test, including paired t-test. Therefore, we provided both Mann-Whitney U test and Wilcoxon signed rank sum test results. 

Table 3. Summary (median (IQR) of indoor air quality during each intervention period (control  and experimental groups for phase 1 and phase 2)

Controls 

(Filter off, n=247)

Experiments 

(Filter on, n=224) 

p-value*

p-value**

PM2.5 (μg/m3)

15.3 (10.2–20.5)

8.8 (4.5–14.2)

<0.001

0.0001

CO2 (ppm)

676.7 (533.9–909.6)

639.2 (487.0–802.2)

0.3026

0.2559

Temperature (°C)

29.4 (28.6–30.6)

29.7 (28.8–31.7)

0.6050

0.0088

Relative humidity (%)

33.8 (30.7–38.8)

35.3 (28.6–40.2)

0.8066

0.2800

PEFR(L/min)

372.0 (319.0–430.0)

378.0 (328.0–417.0)

0.6231

0.8804

*: p-value from Mann-Whitney U test ;  **: Wilcoxon signed rank sum test for the difference in weekly median value by id and order of phase (1st, 2nd and 3rd weekly median values of phase 1 and phase 2)

Additionally, to provide clarification, we provided following sentences in the method section: 

We assumed that 

  1. All observations of PM2.5 in the control and experimental groups were independentof each other. 
  2. The PM2.5 measurements were ordinal
  3. There was no relationship between weekly PM2.5 measurements of control and that of the  experimental group.
  4. In addition, there was no relationship between weekly PM2.5 values in both groups. 

(Reference: Wikipedia, https://en.wikipedia.org/wiki/Mann%E2%80%93Whitney_U_test)

7) Table 4: I would expect each participant to serve as their own control in this analysis, but that does seem not to be the case. I note that the discussion states on line 307-308 that “our… study design allowed that each participant served as their own control”. This is however not the case in the presented analyses.

Participants served as their own control subjects. We provided indoor PM2.5 and PEFR values when they were the experimental group as well as their own control group, under our intervention study design, as seen in Figure 3. 

We already mentioned in our previous manuscript that we used the intervention study design and its advantages, as seen below.

“This study design could provide the strongest empirical evidence of treatment efficacy and minimize selection bias and confounders due to unequal distribution of demographic, socioeconomic, or seasonal factors. It is also known that significant statistical test results are readily interpretable [17].”

8)   discussion line 258: “decreased indoor PM2.5 by up to 43%”. What is meant here? Did PM2.5 decrease by 43% on average, or does the 43% relate to the largest decrease observed? The same goes for line 273: ” median PM2.5 could decrease as much as 6.5 ug/m3. What is meant exactly here?

We provided clarity by updating the sentences as seen below:

(Line 258)

“This intervention study showed that the use of HEPA air purifiers decreased indoor PM2.5 levels by up to 43 % at the asthmatic children’s home in terms of median indoor PM2.5 level between the control group (air purifier without a filter) and the experimental group (air purifier with a filter) over each of the 3-week intervention period.

(Line 273)

We found that the difference in indoor median PM2.5 level between the control and experimental groups was 6.5 μg/m3.

9) line 303: here it states the study included 30 children. Throughout the manuscript a total of 26 children is mentioned. What happened here? Similarly, the number of measurements listed in Table 3 suggest that only about 50% of the daily measurements were available. What happened here? The exact number of participants and number of measurements (n aimed for and n available for analysis) needs to be described.

 Thank you for point this out. We have revised this to 26 from 30. We double checked the number of measurements, and all errors have been corrected. 

This manuscript is a resubmission of an earlier submission. The following is a list of the peer review reports and author responses from that submission.

Round 1

Reviewer 1 Report

The manuscript presented interesting research on the influence of air filtering on indoor PM2.5 and respiratory health in asthmatic children. Air filtering was shown to significantly decrease indoor PM2.5 concentration and children's respiratory issues. The manuscript fits the scope of the journal. Though it lacks experimental details and a thorough literature review, I recommend an acceptance after major revision.

Detailed comments:

  • English writing needs improvements.
  • The introduction section needs significant improvements. Line 63-64, the authors stated that 'relatively less is known about its impact on child asthmatics' health', which is not true. Apparently, the authors missed a thorough literature review about PM2.5 and children's asthma, and should thus supplement relative references to avoid misunderstandings to readers. Some examples:
    • Odajima, H., Yamazaki, S., & Nitta, H. (2008). Decline in peak expiratory flow according to hourly short-term concentration of particulate matter in asthmatic children. Inhalation toxicology20(14), 1263-1272.
    • Tang, C. S., Chang, L. T., Lee, H. C., & Chan, C. C. (2007). Effects of personal particulate matter on peak expiratory flow rate of asthmatic children. Science of the total environment382(1), 43-51.
  • Although the authors reported details about IAQ sampling in a separate paper, some brief introduction on the devices used for IAQ sampling should be interpreted in this paper. It includes the precision of the IAQ sampling devices, frequency of calibration, quality control measures, etc., as they are vital to illustrate the robustness of the results.
  • The unit of the PM2.5 μg/m3 was shown in chaos from Line 178. Please check through the paper.
  • Table 3. Given that you only compared two groups (controls and experiments), why the Kruskal-Warris test was used, not the Mann-Whitney U test?
  • Fig 2 should be re-organized to fit into one page. The resolution was too low. You can mark p-values between every two bars. 
  • Were the HEPA filters used in the study brand new ones? What would be the influence of the HEPA usage-extent on the indoor PM2.5 and children's respiratory health? It could be included in the Discussion section.
  • Were the participants aware that whether they were belonging to the control or experiment group? Awareness/blindness would potentially influence health results.
  • For the same participants, were the results significantly different between phase I and II?

Reviewer 2 Report

In this study, the authors evaluated the effects of indoor PM2.5 on asthmatic children’s peak expiratory flow rate (PEFR) values. Thirty children aged between 6 and 18-year-old were randomly allocated to two groups. The daily indoor PM2.5 was intervened by commercial air purifiers with filter on or off. A multilevel model procedure was used to evaluate the impact of usage of air purifier on change of IAQ (indoor air quality) and consecutively on change of PEFR. Results showed that use of HEPA air purifier decreased indoor PM2.5 concentrations level up to 50 %. Although they observed an association between unit increase of PM2.5 concentration and the decrease of lung functions, the margin of this association was relatively small. Some comments are:

  1. There are so many grammar, editorial errors. For example:

 Lines 15-17, “…there have been relatively limited information is available regarding the associations between the indoor PM2.5 level and respiratory health symptoms in Korean child asthmatics.”

Line112, “…FeNO measurements was performed…..”

Line102, “...Those who born as a premature baby…”

Line 116, “….We used measurement of FeNO at baseline only since our main objective of study was evaluate the association between PEFR and indoor air quality.”

  1. Why the study was designed 1st and 2nd phases? What’s the major difference? The study period only lasted two 3-weeks, how about the long-term effect?
  2. Line 214-216, the author wrote “Larger PM2.5 decrease were observed on the first phase after HEPA deployment in children’s indoor residence, compared with the second and third phase of intervention (Figure 2).”, however, no third phase can be found in the figure.

Reviewer 3 Report

First of all, thank you for the effort in producing this study. This manuscript evaluates the impact of using indoor air filtration on bedroom’s indoor air pollution levels and on its association with Peak Expiratory Flow Rate (PEFR) in asthmatic children.

Despite the relevance and the soundness of this topic, this paper has major flaws, which led the reviewer to conclude that this paper is not suitable for publishing in this journal.

This paper has a seriously poor English spelling, with many grammar errors and spelling faults everywhere. The reviewer also found a repeated phrase (in lines 106-107 and 110-11). Authors should have performed a professional English revision prior to submission.

The introduction is well conducted, although the objective of this paper is not clear.

Most of the major scientific flaws are in the Materials and Methods. Authors did not detailed the methods used, neither for PEFR and FeNO measurements, nor for indoor air pollution monitoring.

In fact, authors followed an outdated international guideline for conducting the spirometry (2005). GINA and ERS/ATS have more recent guidelines on spirometry for young children which should be followed. It is also not clear if there was an experienced operator performing the spirometry, or if the participants performed it by themselves, and how did they validated the data obtained as they were using a portable spirometer. It is also not clear where did authors obtained IgE personal levels.

Authors used low-cost sensors for indoor air pollution monitoring. Besides there was already a prior publication on their calibration, authors should have presented in this paper their calibration and validation in this study conditions. Giving their importance to the quality of the collected data (which was also pointed by the authors in the discussion), these low-cost sensors should be validated and/or calibrated in similar conditions as those found in this study. In fact, measured PM2.5 concentrations seems very low (lower than the recommended WHO limit value). Under this low concentrations, does it make sense to perform an intervention? Are there any associations between the baseline concentrations (without any filter) and the respiratory health of children (namely PEFR of asthmatics)?

Children’s exposure was not accurately estimated. Authors should have used a microenvironmental modelling approach, by considering the concentrations at the time spent in each indoor home microenvironment. In fact, with those low PM2.5 concentrations, it is expected a very low inhaled dose (which also depends on the body weight and on the inhalation rate – low as children will be sleeping in the bedroom) – thus probably not associated with a detectable PEFR increase.

The number of participants is very limited (15). Moreover, it is not entirely clear who are the case and the control group, as authors initially mentioned that everyone was his/her own control, but then they mentioned the existence of a case and a control group (e.g.: line 162).

Results from the models were not entirely clear. Authors should have presented more information, like which covariates the models were adjusted for. Authors also missed model validation. Despite not presenting much information, it seems that the models failed to be adjusted for relevant potential confounders of association, like exposure to other indoor air pollutants, outdoor air pollution, contact with pet animals in the house, existence of carpets in the child’s bedroom, moulds, and so on. Authors should have at least discussed this.

Given the great amount of potential bias in this study, results from the sensitivity analysis should have been presented.

The results from this paper do not provide enough evidence to conclude what authors have claimed in the final paragraph of the discussion.

Conclusion section does not describe correctly the major findings of this study.

Reviewer 4 Report

This is an intervention study. Therefore, the authors should register their studies approved public trials registry. An IRB statement is not a substitute for an approved clinical trial registration.